# Density Functional Theory of Coulombic Excited States Based on Nodal Variational Principle †

**Ágnes Nagy** 

Department of Theoretical Physics, University of Debrecen, H-4002 Debrecen, Hungary; anagy@phys.unideb.hu
† Dedicated to Professor Karlheinz Schwarz on the occasion of his 80th birthday.

**Abstract:** The density functional theory developed earlier for Coulombic excited states is reconsidered using the nodal variational principle. It is much easier to solve the Kohn–Sham equations, because only the correct number of nodes of the orbitals should be insured instead of the orthogonality.

**Keywords:** density functional theory; Coulomb systems; excited states; nodal variational principle

## 1. Introduction

The density functional theory (DFT) [1,2] has been originally worked out for the ground state. It has rigorously been extended to excited states by Theophilou [3] and later by Gross, Oliveira, and Kohn [4–6]. For further extensions and applications of these subspace and ensemble theories, see reference [7]. Subsequently, theories for individual excited states were presented [8–15]. Several works on excited states have been done within the local potential framework [16–29]. Recently, a comprehensive theory for Coulombic excited states has been put forward in a series of papers [30–32]. It takes advantage of the fact that the Coulomb density determines not only its Hamiltonian but the degree of excitation as well and consequently, there is a universal functional valid for any excited state. In addition, excited state Kohn–Sham (KS) equations similar to the ground-state KS equations can be derived.

Recently, Zahariev, Gordon, and Levy [33] have presented a nodal variational principle for excited states. They have proved that the minimum of the energy expectation value of trial wave functions that are analytically well behaved and have nodes of the exact wave function is the exact eigenvalue. This minimum is achieved at the exact eigenfunction.

In this paper, the Coulombic excited state theory is reconsidered utilizing the nodal variational principle. Certainly, the functionals are the same as in the original theory, but it is much easier to solve the Kohn–Sham equations, because only the correct number of nodes of the orbitals should be insured instead of the orthogonality. It is especially important in case of highly excited orbitals.

The paper is organized as follows. In Section 2, the DFT for Coulombic excited states [30–32] is reworked. Section 3 is dedicated to the discussion.

## 2. Coulombic Excited State Theory Using Nodal Variation Principle

The theory is valid for Coulomb external potential $v^{Coul}$. The Hamiltonian has the form

$$\hat{H} = \hat{T} + \hat{V}_{ee} + \sum_{i=1}^{N} v^{Coul}(\mathbf{r}_i)\,, \tag{1}$$

where $\hat{T}$ and $\hat{V}_{ee}$ are the kinetic energy and the electron–electron energy operators. $N$ is the number of electrons and

$$v^{Coul}(\mathbf{r}) = -\sum_{\beta=1}^{M} \frac{Z_\beta}{r_\beta}. \tag{2}$$

$M$ is the number of nuclei and $r_\beta = |\mathbf{r} - \mathbf{R}_\beta|$. $\mathbf{R}_\beta$ and $Z_\beta$ denote the position and the charge of the nucleus $\beta$. Kato's theorem [34–40]

$$\left. \frac{\partial \bar{n}_\beta(r_\beta)}{\partial r_\beta} \right|_{r_\beta=0} = -2Z_\beta n(\mathbf{r} = \mathbf{R}_\beta) \tag{3}$$

is valid both for the ground and any excited state. It has the consequence that the cusps of the density $n$ exhibit the atomic numbers and the positions of the nuclei. In addition, $N$ is given by the the integral of $n$. Hence, $n$ specifies all parameters of the Coulomb potential (2), thus determines the external potential, the Hamiltonian (1), and all properties of the Coulomb system. Furthermore, $n$ cannot be the density for any other Coulomb external potential, that is, two different excited states cannot have the same electron density [30]. Therefore, we might think that the expression

$$F^{Coul}[n] = E[n] - \int n(\mathbf{r}) v^{Coul}[n; \mathbf{r}] d\mathbf{r} \tag{4}$$

would be the appropriate functional for Coulombic densities. However, it is not known how to decide if a density is Coulombic or not. Therefore, instead of (4) $F$ is defined in another way: it is defined for all electron densities not only for Coulombic densities.

As a first step consider a bifunctional

$$F[n, n^{Coul}] = \min_{\substack{\Psi \to n \\ \{\langle \Psi | \Psi_l^{Coul}[n^{Coul}] \rangle = 0\}_{l=1}^{k-1}}} \langle \Psi | \hat{T} + \hat{V}_{ee} | \Psi \rangle , \tag{5}$$

where the minimum is searched over the wave functions that provide the excited state density $n$ and is orthogonal to the first $k-1$ eigenfunctions of the Coulomb system of $n^{Coul}$.

Using the nodal variation principle instead of Equation (5) we can write

$$F[n, n^{Coul}] = \min_{\substack{\Psi \to n \\ \{\Psi \text{ has the nodes of the exact wave function}\}}} \langle \Psi | \hat{T} + \hat{V}_{ee} | \Psi \rangle . \tag{6}$$

It is assumed that a Coulomb density close to $n$ exists.

$$F_\epsilon^{Coul}[n] = \min_{n^{Coul}} F[n, n^{Coul}]; \quad ||n^{Coul} - n|| \leq \epsilon. \tag{7}$$

The smallest $F$ is taken, if there are more than one Coulomb density at the same distance from $n$:

$$F^{Coul}[n] = F_{\epsilon_{min}}^{Coul}[n]. \tag{8}$$

To measure the distance a Sobolev-type norm is applied:

$$d(n^{Coul}, n) \equiv \int \left| \sqrt{n^{Coul}(\mathbf{r}) - n(\mathbf{r})} \right|^2 d\mathbf{r} + \int \left| \nabla \sqrt{n^{Coul}(\mathbf{r}) - n(\mathbf{r})} \right|^2 d\mathbf{r}. \tag{9}$$

The Euler equation is obtained by functional derivation

$$v^{Coul}([n], \mathbf{r}) = -\frac{\delta F^{Coul}[n]}{\delta n(\mathbf{r})} \tag{10}$$

up to a constant.

It is worth emphasizing that the theory above is based on the following statements:

(a) The cusps and the asymptotic decay of the Coulombic density determine the external potential and the ionization potential;

(b) It is supposed that bifunctional $F[n, n^{Coul}]$ (Equation (5) or (6)) exists, where $n^{Coul}$ is close to $n$. Further, the existence of $F^{Coul}[n]$ (defined by the Equations (7) and (8)) is assumed;

(c) Equation (6) is based on the assumption that the nodes of the exact excited state wave functions are known;

(d) It is assumed that the functional derivative of $F^{Coul}[n]$ exists. It is needed to derive the Euler Equation (10).

Consider now the Kohn–Sham (KS) system. In our original definition the non-interacting kinetic energy bifunctional was written

$$T_s^{Coul}[n, n^{Coul}] = \min_{\substack{\Phi \to n \\ \{\langle \Phi | \Phi_l[n^{Coul}] \rangle = 0\}_{l=1}^{k-1} \\ ||n_1^{Coul} - n_1^0|| \leq \delta}} \langle \Phi | \hat{T} | \Phi \rangle, \tag{11}$$

where the search is over the wave functions $\Phi$ having the excited state density $n$ and orthogonal to the first $l - 1$ eigen functions of the non-interacting system. The excited state density is the same in the real and the KS systems. If there are more than one KS system with the same density $n^{Coul}$, the one closest to the true ground-state density $n_1^{Coul}$ is taken. Instead of Equation (11) we can write

$$T_s^{Coul}[n, n^{Coul}] = \min_{\substack{\Phi \to n \\ \{\Phi \text{ has the nodes of the exact wave function}\} \\ ||n_1^{Coul} - n_1^0|| \leq \delta}} \langle \Phi | \hat{T} | \Phi \rangle \tag{12}$$

using the nodal variation principle. The existence of a unique Coulomb density close to the non-Coulomb density $n$ is assumed:

$$T_{s,\epsilon}^{Coul}[n] = \min_{n^{Coul}} T_s[n, n^{Coul}]; \quad ||n^{Coul} - n|| \leq \epsilon. \tag{13}$$

It is supposed that there is at least one Coulomb density closer to $n$ than $\epsilon$, provided that $\epsilon$ is large enough. The minimum specifies the final form:

$$T_s^{Coul}[n] = T_{s,\epsilon_{min}}^{Coul}[n]. \tag{14}$$

The functional derivation yields an Euler equation, within an additive constant,

$$w^{Coul}([n], \mathbf{r}) = -\frac{\delta T_s^{Coul}[n]}{\delta n(\mathbf{r})}. \tag{15}$$

The KS theory presented above is based on the following statements:

(a) The existence of the non-interacting kinetic energy bifunctional $T_s^{Coul}[n, n^{Coul}]$ (Equation (11) or (12)) with $n^{Coul}$ close to $n$ is assumed. Further, it is presumed that $T_s^{Coul}[n]$ constructed by Equations (13) and (14) exists;

(b) Equation (12) is based on the assumption that the nodes of the non-interacting excited state wave functions are known;

(c) It is supposed that the functional derivative $T_s^{Coul}[n]$ exists and the Euler Equation (15) can be derived.

It is convenient to partition $F^{Coul}[n]$ as

$$F^{Coul}[n] = T_s^{Coul}[n] + J^{Coul}[n] + E_{xc}^{Coul}[n], \tag{16}$$

where $J^{Coul}[n]$ and $E_{xc}^{Coul}[n]$ are the classical Coulomb and exchange-correlation energies. Equations (10), (15) and (16) lead to the KS potential

$$w^{Coul}([n], \mathbf{r}) = v^{Coul}([n], \mathbf{r}) + v_J^{Coul}([n], \mathbf{r}) + v_{xc}^{Coul}([n], \mathbf{r}) \tag{17}$$

as the sum of the external, the classical Coulomb and the exchange-correlation potentials. The density has the form

$$n = \sum_{i=1}^{K} \lambda_i |\phi_i|^2, \tag{18}$$

where the KS orbitals $\phi_i$ are solutions of the KS equations

$$\left[ -\frac{1}{2}\nabla^2 + w^{Coul}([n], \mathbf{r}) \right] \phi_i = \varepsilon_i \phi_i. \tag{19}$$

The occupation numbers $\lambda_i$ are 0, 1, or 2 for a non-degenerate system. $K$ denotes the orbital having the highest orbital energy with non-zero occupation number.

## 3. Discussion

In the present version of the Coulombic excited state theory, the variation is done over the trial wave functions having the nodes of the exact wave functions both in the interacting and the non-interacting systems. That is, the sole difference between the original and the present forms of the theory is using Equations (6) and (12), instead of Equations (5) and (11). Despite this difference, the functionals are the same as in both versions. Generally, the nodes are not known. The wave functions are not known either. In DFT we define functional $F[n]$ via the wave function, but we do not actually use this definition in calculations. Only, $F$ as a functional of $n$ is applied.

On the other hand, in DFT the exact functionals are not known and approximate functionals are applied in calculations. Additionally, in actual calculations the KS Equations (19) are solved. The nodal variational principle leads to a huge simplification, inducing much easier calculations. It is the consequence of the fact that the variational problem reduces to the solution of the KS equations. The orbitals, that is, one-particle functions have to be obtained. If the electron configuration of the state is known, we have to solve the KS equations insuring either the orthogonality of orbitals or the correct number of nodes of the orbitals. The latter is simpler as it is explained in the example below. Certainly, we have to know the correct number of nodes of the orbitals.

The nodal behavior of eigenfunctions were discussed in several papers (see, e.g., [41–44]). Still the number of nodal surfaces is rarely counted in calculations. Hatano and coworkers [43,44] developed a computer program to count the number of nodal regions and applied it in molecular orbital calculations.

Recently, the original Coulombic excited state theory [30–32] has been discussed [7]. The localized Hartree–Fock (LHF) [45] and the Krieger, Li, and Iafrate (KLI) [46] methods combined with correlation have been generalized for excited states. In addition, several highly excited states of Li and Na atoms have been studied.

The radial KS equations can be solved using Numerov's algorithm [47] searching eigenvalues with the correct number of nodes. This method was used by Herman and Skillman in their Hartree–Fock–Slater computer code [48]. We do not have to check the orthogonality of the orbitals during the calculations, only the number of nodes has to be counted. The correct number of nodes is enough to insure the orthogonality. It is especially beneficial in studying higher excited states. In [7], several highly excited states of Li and Na atoms have been studied. Calculations have been performed with KLI and KLI plus (local Wigner) correlation (see details in [7]). Take, for example, the configuration $1s^2 5s$. The orbital $\phi_{5s}$ should be orthogonal to all the orbitals below, that is, $\phi_{1s}$, $\phi_{2s}$, $\phi_{3s}$, and $\phi_{4s}$. The orbitals $\phi_{2s}$, $\phi_{3s}$, and $\phi_{4s}$ have zero occupation numbers, do not contribute to the density, so we do not have to calculate them. It is enough to calculate the orbital $\phi_{5s}$ and the correct number of nodes insures the orthogonality to all the orbitals below. We emphasize that as the configuration $(1s^2 5s)$ is known, we know the exact number of nodes of the radial orbitals. The radial orbital $\phi_{ks}$ has $k-1$ nodes. (Because of the spherical symmetry of the system, the radial KS equations should be solved.) We can easily check numerically that the

orbitals with the correct number of nodes are really orthogonal. We calculated the orbitals $\phi_{2s}$, $\phi_{3s}$, and $\phi_{4s}$, and the integrals $\int \phi_{k_1} \phi_{k_2} d\mathbf{r}$, where $k_1$ and $k_2$ can be $1s, \ldots, 5s$. We found that the absolute value of the integral was always less than $10^{-6}$ for $k_1 \neq k_2$.

In summary, the Coulombic excited state theory has been re-examined based on the nodal variational principle. The functionals are the same as in the original theory, but the solution of the Kohn–Sham equation is much easier as only the correct number of nodes of the orbitals should be insured instead of the orthogonality.

**Funding:** This research was supported by the National Research, Development and Innovation Fund of Hungary, financed under 123988 funding scheme.

**Institutional Review Board Statement:** Not applicable.

**Informed Consent Statement:** Not applicable.

**Data Availability Statement:** Data sharing is not applicable to this article.

**Conflicts of Interest:** The author declares no conflict of interest.

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
