# Peer review of "Density Functional Theory of Coulombic Excited States Based on Nodal Variational Principle"

_computation, doi:10.3390/computation9080093_

Round 1

Reviewer 1 Report

The author is a world renounced expert on this topic and this theoretical work is a continuation of many of her excellent theoretical developments in density functional theory. The manuscript is clear, novel, easy to follow and well written. To accept as is is my recommendation.

Author Response

The author is a world renounced expert on this topic and this theoretical work
 is a continuation of many of her excellent theoretical developments in 
density functional theory. The manuscript is clear, novel, easy to follow and
 well written. To accept as is is my recommendation.

There were no questions.

Reviewer 2 Report

Please see my attached Report.

Author Response

The article presents an elegant combination of the recently formulated nodal
 variational principle for excited states with a Kohn-Sham approach to excited
 states based on the rigorous Coulombic density functional formalism developed by the author with collaborators. The possible implications of the presented  approach are both conceptual and practical. It would be interesting to find  if one can utilize any known properties of Kohn-Sham orbital nodes in the presented approach.

Answer:

The present approach can be utilized in the solution of KS equation. It is
especially useful for highly excited states. More detail are added on
the example  shown in Discussion.

Reviewer 3 Report

The article presents an elegant combination of the recently formulated nodal variational principle for excited states with a Kohn-Sham approach to excited states based on the rigorous Coulombic density functional formalism developed by the author with collaborators. The possible implications of the presented approach are both conceptual and practical. It would be interesting to find if one can utilize any known properties of Kohn-Sham orbital nodes in the presented approach. 

Round 2

Reviewer 2 Report

The author has made the requisite changes.